# Proteomic Analyses Detect Higher Expression of C-Type Lectins in Imidacloprid-Resistant Colorado Potato Beetle *Leptinotarsa decemlineata* Say

**DOI:** 10.3390/insects12010003

**Published:** 2020-12-23

**Authors:** Ian M. Scott, Gabrielle Hatten, Yazel Tuncer, Victoria C. Clarke, Kristina Jurcic, Ken K.-C. Yeung

**Affiliations:** 1London Research and Development Centre, Agriculture and Agri-Food Canada, London ON N5V 4T3, Canada; ghatten@uwo.ca (G.H.); ytuncer@uwo.ca (Y.T.); 2London Regional Proteomics Centre, Biochemistry, Western University, London ON N6A 5C1, Canada; vwraw@uwo.ca (V.C.C.); kjurcic@uwo.ca (K.J.); kyeung@uwo.ca (K.K.-C.Y.)

**Keywords:** proteomics, insecticide resistance, Colorado potato beetle, detoxification enzymes, C-type lectins

## Abstract

**Simple Summary:**

Surveillance and determining the mechanisms of pesticide resistance are key components of resistance management. Mechanisms can be investigated using biochemical, genomic, proteomic and other modern analytical techniques. In the present study, proteomic analyses of Colorado potato beetle (CPB), one of the most adaptable insect pests to both plant toxins and synthetic insecticides, were applied to identify protein differences in insecticide-susceptible and resistant strains. Proteins identified in abdominal and midgut tissues based on separating by 2-dimensional (2-D) gels and mass spectrometry were associated with insect innate immunity. A database search found that the highest match was a C-type lectin (CTL), which is a component in the insect’s innate immune system. The 2-D gel spot identified as a CTL was greater in the insecticide-resistant CPB strain, but the CTL spot size was increased by exposure to imidacloprid in the susceptible strain. This is a novel finding, which suggests that CTLs and insect immunity may respond to certain toxins as well as to pathogens. There may also be a potential application for pest management if insect immunity is targeted.

**Abstract:**

The Colorado potato beetle (CPB) is one of the most adaptable insect pests to both plant toxins and synthetic insecticides. Resistance in CPB is reported for over 50 classes of insecticides, and mechanisms of insecticide-resistance include enhanced detoxification enzymes, ABC transporters and target site mutations. Adaptation to insecticides is also associated with changes in behaviour, energy metabolism and other physiological processes seemingly unrelated to resistance but partially explained through genomic analyses. In the present study, in place of genomics, we applied 2-dimensional (2-D) gel and mass spectrometry to investigate protein differences in abdominal and midgut tissue of insecticide-susceptible (S) and -resistant (R) CPB. The proteomic analyses measured constitutive differences in several proteins, but the highest match was identified as a C-type lectin (CTL), a component of innate immunity in insects. The constitutive expression of the CTL was greater in the multi-resistant (LI) strain, and the same spot was measured in both midgut and abdominal tissue. Exposure to the neonicotinoid insecticide, imidacloprid, increased the CTL spot found in the midgut but not in the abdominal tissue of the laboratory (Lab) strain. No increase in protein levels in the midgut tissue was observed in the LI or a field strain (NB) tolerant to neonicotinoids. With the exception of biopesticides, such as *Bacillus thuringiensis* (Bt), no previous studies have documented differences in the immune response by CTLs in insects exposed to synthetic insecticides or the fitness costs associated with expression levels of immune-related genes in insecticide-resistant strains. This study demonstrates again how CPB has been successful at adapting to insecticides, plant defenses as well as pathogens.

## 1. Introduction

The Colorado potato beetle (CPB) *Leptinotarsa decemlineata* Say (Coleoptera: Chrysomelidae), is a major insect pest of potato, *Solanum tuberosum* L. (Solanaceae), as it adapted from feeding on wild relatives to commercial potato varieties and has developed resistance to many insecticide classes as well as plant defenses [1]. A commonly used insecticide in the past two decades has been the neonicotinoid, imidacloprid, and a range of resistance and cross-resistance has developed in CPB populations in several potato growing regions [2,3,4,5]. The CPB has adapted to many classes of insecticides registered for its control through increased expression of detoxification enzymes (P450s, GSTs, esterases), ABC transporters, altered target sites (genetic mutations) or a combination [2,6,7]. Resistance can lead to different fitness costs in CPB [3,8,9], possibly due to the requirement for maintaining higher levels of metabolic or detoxification enzymes. There is evidence that insecticide resistance increases the energy needed to produce the metabolic enzymes required to detoxify the chemicals [10,11,12], altered fatty acid metabolism [13], and has been associated with behaviour changes [14]. CPB can also adapt to plant defenses, such as proteinase inhibitors, that negatively affect serine and cysteine digestive peptidases and reduce protein digestion [15] by changing the gut composition to avoid the plant’s defense mechanism [16]. The explanation for the success of members of the Chrysomelidae is thought to be the differential susceptibility to plant peptidase inhibitors, in part through gene duplication and selection for inhibitor insensitive genotypes [15].

To determine the mechanisms of resistance and adaptation in insects, several approaches, including measurement of biochemical and genetic differences, as well as alterations in protein expression, are often applied. In the first 2 cases, the techniques may only indicate a general, rather than specific, mechanism of resistance. The recent sequencing of the CPB genome has provided evidence of the wide range of gene expansion and elevated expression of digestive enzymes in gut tissues [15]. However, measurements at the proteome level can provide additional information about the functioning of cells and complement the transcriptomic data to define critical biological functions in insects [17]. The application of a proteomic approach includes 2-D polyacrylamide gel electrophoresis (2D PAGE) to separate proteins by mass and charge, followed by visualization using either fluorescent or visible range stains. This allows for a direct comparison between paired samples, highlighting any differential protein expression, which can later be identified through Matrix Assisted Laser Desorption/Ionization (MALDI) Mass Spectrometry (MS) [18] or Liquid Chromatography (LC) Electrospray (ESI) Tandem MS [19,20]. The use of MS for the detection of peptides greatly improved the capacity for comprehensive analysis of protein expression [21]. MALDI-MS is the preferred method of protein mass spectrometry due to high throughput, straightforwardness, tolerance to contaminants, such as salt, and giving a direct relation between the m/z value and actual peptide mass [22,23]. Tandem mass spectrometry has been used to sequence peptides and proteins of interest in insects, such as the green peach aphid [24], the brown planthopper [25], and the Asian corn borer [26]. Proteomic analysis is particularly useful in the case of insect species where the genome has not yet been sequenced, as the integration of proteomic and available transcriptomic data can enhance the characterization of biological pathways and networks involved in controlling traits [27].

To date, few proteomic studies have been applied in the study of insecticide resistance in CPB. The objective of this study was to examine the proteome of a laboratory reference strain and a multi-resistant strain of CPB regularly selected for resistance to neonicotinoids. Differential protein levels were then compared in a field-collected CPB population identified as tolerant to neonicotinoid insecticides.

## 2. Materials and Methods

### 2.1. CPB Strains

An insecticide-susceptible laboratory (Lab) strain was reared at Agriculture and Agri-food Canada (AAFC), London, ON, Canada, for >100 generations. A multi-resistant strain, originally collected in Long Island (LI), NY, USA, was obtained from Dr. Z. Szendrei, Department of Entomology, Michigan State University (MSU), East Lansing, MI, USA. A field-collected CPB population was obtained from potato fields in New Brunswick (NB), Canada. All strains were reared under identical conditions, fed on greenhouse-grown Kennebec variety potato (*Solanum tuberosum)* at 25 °C, 50% relative humidity, and 16:8 L:D photoperiod.

### 2.2. Insecticide Exposure

CPB adults were treated with 1 µL acetone or 1 µL of a solution of technical imidacloprid in acetone at a range of 5 to 6 imidacloprid concentrations (10–150 μg/mL for susceptible, 0.1–2.5 mg/mL for tolerant, 1–17.5 mg/mL for resistant) for 3 strains (Lab, NB and LI). Insecticide bioassays were 7 days in length and followed the methods developed in the laboratory where a minimum of 3 replicates bioassays were performed with 10 adults per concentration [3]. Probit analysis of the 7 day mortality data generated was used to determine median lethal dose (LD_50_) values, fiducial limits and resistance ratios (RRs) relative to the laboratory strain for the LI and NB strains. Adult CPB of the Lab, LI and NB strains were treated with sublethal doses (10% of the LD_50_, or 10 µg/mL and 750 µg/mL, respectively) of insecticide on the ventral side of the abdomen. A control group was left untreated. The CPB were provided with potato leaves, and after 48 h, the insects were dissected to remove tissues (midgut and abdominal contents minus the midgut) to be used in detoxification enzyme and proteomic analyses.

### 2.3. Cytochrome P450 and b5 Activity

The analysis of CPB monooxygenase enzyme activity modified the methods developed for housefly [28]. Briefly, twenty CPB 4th instar larval midguts and fat bodies were dissected, and the gut food bolus was removed. The collected tissues were combined in 1 mL of homogenization medium (HM) composed of 40% glycerol (50 mL), 1 mL 100 mM ethylene diamine tetra acetic acid (EDTA) (1 mL), 10 nM dithiothreitol (DTT) (1 mL), 100 mM phenylmethyl sulfonylfluoride (PMSF) (1 mL), 100 mM phenylthiourea (PTU) (1 mL), and 0.1 M, pH 7.5 sodium phosphate (NaP) buffer (71 mL). The tissue solution was homogenized at 1000 rpm for 1 min and centrifuged at 10,000× *g* for 20 min at 4 °C. The supernatant was collected and centrifuged at 100,000× *g* for 1 h at 4 °C. The pellet was re-dissolved in a 2 mL re-suspension medium (RM) composed of 40% glycerol (50 mL), 100 mM EDTA (1 mL), 10 mM DTT (1 mL), 100 mM PMSF (1 mL), and 0.1 M, pH 7.5 NaP buffer (47 mL). The microsome homogenate solution was stored at −80 °C until enzyme activity was performed. Cytochrome b5 and P450 activity were determined following the methods described for housefly [29] using a spectrophotometer (Beckman DU-600). Microsome protein was determined by the Bradford method [30]. From each microsome sample, 0.2 mg was added to a quartz cuvette using a 100 µL micropipette, followed by phosphate buffer to a final volume of 0.4 mL. The cuvette was placed in the spectrophotometer and scanned between 400 and 500 nm and zeroed as the blank. A small amount (mg range) of sodium hydrosulfite was added to the cuvette, inverted with parafilm covering the opening to mix, and then re-inserted in the spectrophotometer to read the b5 peak. Immediately afterwards, the dithionite-reduced microsomes were zeroed as the blank carbon monoxide was bubbled into the cuvette for 30 s, followed by a scan for the P450 peak. Cytochrome b5 activity (nmoles × mg protein^−1^) was calculated by subtracting the absorbance at 409 nm from the maximum absorbance wavelength (approx. 426 nm) and dividing by the extinction coefficient for b5, 184 cm^−1^ mM^−1^. Cytochrome P450 activity (nmoles × mg protein^−1^) was determined by subtracting the absorbance at 490 nm from the absorbance at 450 nm and dividing the P450 extinction coefficient, 0.091 cm^−1^ mM^−1^. The difference in cytochrome P450 and b5 activity between CPB strains, pre- and post-treatment with imidacloprid, was determined by two-way ANOVA and Tukey’s HSD comparison of means (2001; SAS Institute, Cary, NC, USA).

### 2.4. Proteomic Analysis

Midgut and abdominal tissue samples collected from imidacloprid and control CPB were flash-frozen in liquid nitrogen and stored at −80 °C. Further processing was completed at the Functional Proteomics Facility (FPF), University of Western Ontario, London.

Tissue samples were homogenized with a pestle in lysis buffer (8 M urea, 2% CHAPS, 40 mM dithiothreitol (DTT), and Protease Inhibitor mix (Amersham Biosciences, Piscataway, NJ, USA), then sonicated. Proteins were extracted using the 2D Clean-Up Kit (Amersham Biosciences). Protein samples were frozen at −80 °C for later use, and concentrations were determined using a Bradford assay [30]. The proteins were separated on pH 3–10 NL, 7 cm gel strips (Immobilien Dry strips, Cytiva, Marlborough, MA, USA). Rehydration buffer containing pH 3–10 NL IPG buffer was mixed with cell lysate (100 µg of protein) to a total volume of 125 μL and was added to the strip holder boat. The strip was placed in the boat and covered with 400 μL of Dry Strip Cover Fluid (Cytiva). The gel was run with the Ettan IPGphor for the first dimension run with a total run time of 22 h. Strips were removed from the boat and equilibrated with 0.05 g DTT in 5 mL SDS equilibration buffer followed by equilibration with 0.125 g Iodoacetamide (IAA). Gels were rocked with gel side up for 15 min in each solution. Gel strips and an SDS polyacrylamide standard (5 µL of BioRad unstained protein standard was added to a 4 mm × 4 mm piece of filter paper and allowed to dry) was added to the SDS polyacrylamide gel and sealed with agarose sealing mixture. The second dimension gel was run at 200 V for about 1 h, or until the dye front was just about to run off the gel. The gels were stained with SYPRO Ruby gel stain for high-resolution fluorescent imaging and then with GelCode Blue for visualization by eye. The 1st dimension step was further optimized by using larger (13 cm and 18 cm) gels for greater spot separation and able to load and extract a higher concentration of protein (up to 250 µg).

### 2.5. Imaging and Mass Spectrometric Analysis

Gels were imaged using the PerkinElmer ProXPRESS 2D Proteomic Imaging System (PerkinElmer, Waltham, MA, USA) at the FPF, UWO. Spots of interest were determined by eye from the overlapping gel after staining with GelCode Blue stain followed by using Progenesis SameSpots software (Nonlinear Dynamics, Newcastle, UK) with a one-way ANOVA (significance level of *p* < 0.05). Spot selection used an Ettan Spot-Picker followed by protein digestion with the MassPrep automated station (PerkinElmer, Inc., Waltham, MA, USA). Gel pieces were Coomassie destained using 50 mM ammonium bicarbonate and 50% acetonitrile, which was followed by protein reduction using 10 mM dithiotreitol (DTT), alkylation using 55 mM iodoacetamide (IAA), and tryptic, chymotryptic, or AspN digestion. Peptides were extracted using a solution of 1% formic acid and 2% acetonitrile and lyophilized. Before mass spectrometry analysis, dried peptide samples were re-dissolved in a 10% acetonitrile and 0.1% TFA (trifluoroacetic acid) for MALDI mass spectrometric analysis (Sciex 4700 MALDI TOF/TOF, Analyzer, Framingham, MA, USA) or in 0.2% formic acid for ESI LC-MS/MS analysis (Waters QTOF Global, Milford, MA, USA). The MALDI MS instrument was equipped with a 355 nm Nd:YAG laser at 200 Hz. Reflectron positive ionization MS and MS/MS modes were used. The QTOF instrument was equipped with a Z-spray (ESI) source. Samples were run in positive ion mode and using Data Dependent Acquisition. A reversed phase type liquid chromatography (LC) column was coupled to the mass spectrometer.

### 2.6. Protein Identification

QTOF raw data were searched using PEAKS Studio 10.5 (build 20,200,219). The Potato Beetle Database was used for a search (NCBI, 11,399 entries). Parent mass error tolerance and fragment mass error tolerance were set to 0.2 Da. FDR was estimated using a Target-Decoy. Carbamidomethylation (C) was selected as a fixed modification, while deamidation (NQ) and oxidation (M) were selected as variable modifications, with maximum 5 variable modifications per peptide allowed. Three missed cleavages were selected for tryptic, and four missed cleavages were selected for chymotryptic and AspN digests.

In addition, three peptide mass fingerprint (PMF) MALDI MS data were searched using Mascot, ProFound, and Protein Prospector search engines. The NCBInr, MSDB, SwissProt and UniProtKB databases were used with no restriction on taxonomy. Estimated protein molecular weights and isoelectric points from the gels were used to evaluate matches further.

## 3. Results

### 3.1. Imidacloprid Susceptibility

The CPB adults treated with imidacloprid were found to differ in mortality over the course of 7 days. The Lab strain had the lowest LD_50_ compared to the NB and LI strains (Figure 1), with a resistance ratio of 6.1 and 92.3, respectively, relative to the Lab strain value (mortality data was from our previous study [4]).

### 3.2. Cytochrome P450 and b5 Activity

There was a significant difference in P450 activity between susceptible (Lab) strain CPB pre- and post-imidacloprid exposure (Two-Way ANOVA; d.f. = 2.12; F = 5.12; *p* = 0.0247), but no difference was determined between the Lab strain and the NB or LI strains (Tukey’s test, *p* > 0.05) with untreated microsomes (Figure 2A) or imidacloprid-treated microsomes (Figure 2B).

Similarly, the cytochrome b5 activity increased for the Lab strain microsomes pre- to post-imidacloprid exposure (Two-Way ANOVA; d.f. = 2.12; F = 4.12; *p* = 0.0434), but no difference was determined between the Lab strain and the NB or LI strains (Tukey’s test, *p* > 0.05) with untreated microsomes (Figure 3A) or imidacloprid-treated microsomes (Figure 3B).

### 3.3. Proteomic Analyses

#### 3.3.1. Abdominal Contents

The most prominent spot of interest (IS25) found in tissues removed from the abdomen (minus the midgut) 2D PAGE strips (pH 4–7, 13 cm) was found in the NB and LI sample, but not in the Lab strain (Figure 4). The protein of interest appears to be of the same molecular weight (MW) and isoelectric point (pI) in tissues from both abdominal contents and the midgut. The spot was digested and analyzed by ESI LC-MS/MS, where the peptide fingerprint database searches found a protein identified as ladderlectin-like, or C-type lectin, from *L. decemlineata* with a high match (62% coverage) (Table 1). The next highest match was an E-selectin-like, but only half the percent match of the lectin protein.

#### 3.3.2. Midgut

Of the many 2-D gel spots observed on the pH 4–7, 13 and 18 cm strips, the majority had a low association with known proteins. However, two, in particular, held promise: spots IS17 and IS51 (See Figure 5 and Figure 6; Appendix A). In midgut samples from CPB unexposed to imidacloprid, the two proteins were found as dense proteins in LI and more faintly in NB, but not at all in the Lab strain (Figure 5 and Figure 6; Appendix A). However, after exposure to imidacloprid, denser amounts of proteins were observed in the Lab strain gels, whereas the amount of protein declined in the LI and NB samples (Figure 5 and Figure 6; Appendix A).

The best match using ESI LC-MS/MS peptide sequences for IS17 was also a ladderlectin-like protein from *L. decemlineata* (Table 2), the same as spot IS25 from the abdominal contents (Figure 5). The match for IS17 was not as high as the match for IS25 (40% versus 62% coverage). The best match (8%) using ESI LC-MS/MS peptide sequences for IS51 was a digestive cysteine protease intestain from *L. decemlineata* (Table 3).

Spot IS17 was also analyzed using MALDI MS, and the spectra from the LI, Lab and NB strain midgut tissues were compared (Appendix A, respectively). The best match for IS17 from LI using the database searches was again the ladderlectin-like protein from *L. decemlineata* (Appendix A), and the match was the same (40.6% coverage) compared to the match from the ESI LC-MS/MS analyses (Table 2). The same protein was identified for the IS17 from the Lab strain tissue. However, it was much lower (6.6% coverage) (Appendix A).

## 4. Discussion

The present study highlights not only the importance of understanding the underlying mechanisms of insecticide resistance in potato pest species, such as CPB but also the immune response should fitness costs allow greater vulnerability to pathogens. Fitness costs that have been observed in imidacloprid-resistant CPB include reductions in fecundity, such as egg hatch success [8]. Knowing that fitness costs exist in a population allows for the potential use of refuges or the promotion of rotations or other resistance management strategies to maintain the susceptibility to imidacloprid. This was the first report of elevated C-type lectins (CTLs) in imidacloprid-resistant and -tolerant CPB and that exposure to imidacloprid-induced increased levels in the same lectins in insecticide-susceptible CPB. The constitutively higher levels of CTLs in the tolerant and resistant strains, and the induced levels in the susceptible strain, indicate a common response by CPB to the demands of maintaining a higher immune response.

Unlike vertebrates, insects do not have an adaptive immune system and must rely on their innate immune system to protect them [31,32]. A critical initial function of the innate immune system is identifying pathogens. This is performed by pattern recognition receptors (PRRs), which are germline-encoded proteins designed to recognize pathogen-associated molecular patterns (PAMPs) on the surface of pathogens. Examples of PRRs include peptidoglycan recognition proteins (PGRPs), Gram-negative binding proteins (GNBPs), and C-type lectins (CTLs) [31,32].

CTLs are a family of proteins that contain carbohydrate-recognition domains (CRDs). CRDs are typically composed of 110–130 amino acid residues, form a canonical fold containing α-helices, β-sheets, and loops, which are stabilized by two or three pairs of disulfide bonds [31]. CTLs can be divided into three subfamilies based on the number of CRDs they contain [31,32,33]. Most insects have CTLs with only one CRD, categorized as CTL-S. CTLs with two CRDs are part of the immulectin family (IML), and those with other functional domains in addition to their CRD are part of the CTL-X family. The CTL-X family is most commonly found in lepidopteran species [34], and these tandem CRDs allow for a greater range of microorganisms to which it can bind [35]. Binding of the CTL to the pathogen occurs when key hydroxyl groups of the sugar ligands of the pathogen form hydrogen bonds with acidic and amide amino acid side chains on the CTL [33].

The functions of insect CTLs in innate immunity include encapsulation, phagocytosis, opsonization, nodule formation, agglutination, melanisation, and prophenoloxidase to combat viral, nematode, and fungal infection [31,32,33,34,36]. For example, the CTL IML-10 in the Asian corn borer *Ostrinia furnacalis* Guenée (Lepidoptera: Crambidae) is secreted by the fat body into the larval plasma to encapsulate foreign objects [37]. RNAi was used to knockout IML-10, resulting in a decrease in encapsulation from 90% to 30%, indicating IML-10 was partially responsible for the encapsulation process. In another case, the CTL HaCTL3 in the cotton bollworm *Helicoverpa armigera* Hübner (Lepidoptera: Noctudiae) was shown to mediate normal growth and development of *H. armigera* through suppression of *Enterocuccus mundtii* Collins et al. (Lactobacillales: Enterococcaceae), a lactic acid bacteria, in the hemolymph [38]. RNAi of HaCTL3 led to the suppression of ecdysone and juvenile hormone signalling and consequently reduced the larval body size and delayed pupation.

The present study is the first to indicate that CTL protein expression in a coleopteran is greater in imidacloprid-resistant CPB than in insecticide-susceptible. The significance of this finding has yet to be determined, but the discovery of a specific CTL in another coleopteran species is believed to be important for future pest management. A CTL with dual CRDs, TcCTL3, in the red flour beetle *Tribolium castaneum* Herbst (Coleoptera: Tenebrionidae) is located in the central nervous system and hemolymph [33]. When TcCTL3 was knocked down by RNAi, the expression of nine antimicrobial peptides (AMPs) and four transcriptase factors stimulated by lipopolysaccharide (LPS) and peptidoglycan (PGN) were significantly decreased, leading to greater *T. castaneum* mortality when infected with Gram-positive *Staphylococcus aureus* or Gram-negative *Escherichia coli* infection. Since TcCTL3 plays an important role in regulating the immune response of *T. castaneum* through pattern recognition, agglutination, and mediating AMP expression, it could be used as a potential target for pest control strategies.

In two separate studies of CTL in mosquitoes, findings indicate that insecticide resistance results in limited costs to the immune-gene expression in mosquitos, but that biopesticides from Bt can compete with CTL for targets which would affect the activity of the Cry toxins in the insect. The expression of several immune-related genes was measured after stimulation by LPS in insecticide-resistant and -susceptible populations of Lab strains and a sympatric field population with resistant and susceptible individuals of the common house mosquito *Culex pipiens* L. (Diptera: Culicidae) [39]. The insecticide-resistant Lab strain showed a drastic increase in the expression of immune-related genes after stimulation compared to the susceptible Lab strain, while the expression levels of immune-related genes were no different between the resistant and susceptible individuals of the field strain. The CTL CTLGA9 in the yellow fever mosquito *Aedes aegypti* L. (Diptera: Culicidae) was induced by Bt Cry toxins by interacting with brush border membrane vesicles (BBMVs) of the larvae and with alkaline phosphate (ALP1) and aminopeptidase-N (APN), the binding targets of Cry toxins [40]. These results demonstrate that CLTGA9 could compete with Cry toxins for the binding sites of APL1 and APN and thereby inhibit the larvicidal properties of the biopesticide. It is unlikely the CPB C-type lectin identified in this study would interfere with the imidacloprid binding site. However, knockdown of the gene could still increase the susceptibility to pathogens. Proteomic analysis appears to be a method that can be used to separate and identify CTL proteins in the gut tissue of insects to confirm the effect of RNAi.

Proteomic analysis has been used previously to measure changes in insect metabolism and digestion between insecticide-resistant and -susceptible strains or after insecticide exposure. In one case, whole bodies of deltamethrin-resistant 4th instar common house mosquito *Culex pipiens pallens* L. (Diptera: Culicidae) larvae were found to have 30 differently expressed proteins compared to the susceptible strain, with 15 upregulated and 15 downregulated [41]. The most notable change detected was a significant increase in the P450 CYP6AA9, which was 2 to 2.4-fold greater in the resistant compared to the susceptible strain. In the present study, P450 proteins were not detected in the abdominal or midgut tissues of the CPB by MALDI MS or ESI LC-MS/MS, but a biochemical assay did measure total P450 at higher levels in the resistant strain when tissues from 20 insects were combined. There are many P450s present in CPB that are induced by imidacloprid, but individually contribute only in a small way to the overall resistance [6]. Therefore, the proteins associated with induced CYP genes may not be distinguished clearly by 2-D PAGE compared to the proteins associated with metabolism that appear more concentrated. In addition, the detection of CYP6AA9 in the mosquito may be due to the more sensitive approach applied. Peptides were labelled with isobaric tags for relative and absolute quantitation (iTRAQ) analysis, coupled with nano-electrospray ionization followed by tandem mass spectrometry (MS/MS) in high energy collision dissociation operating mode [41]. A similar technique was used to compare protein differences in chlorpyrifos-resistant western flower thrips (WFT) *Frankliniella occidentalis* Pergande (Thysanoptera: Thripidae), where iTRAQ was followed with fractionation using an Ulteremex SCX column and LC-20AB HPLC system [42]. Out of a total of eight, one of those upregulated was a cytochrome P450 CYP6-like protein (1.91-fold higher) in the resistant compared to the susceptible strain. Other differentially expressed proteins in WFT were involved in catalytic and enzyme regulation activity, indicating these processes are likely related to the mechanisms of resistance to chlorpyrifos. Similarly, in silverleaf whitefly, *Bemisia tabaci* Gennadius (Hemiptera: Aleyrodidae), the more resistant biotype B relative to the biotype Q, had an over-expressed carboxylesterase 2 (Coe2) protein, which is believed to contribute to insecticide resistance in this strain [43]. In this case, the samples were loaded into a second stage NuPAGE 4–12% Bis-Tris gel after separation by SDS PAGE, presumably to concentrate proteins further. In contrast, protein differences in the green peach aphid *Myzus persicae* Sulzer (Hemiptera: Aphididae) resistant to imidacloprid were greatest with serine/threonine-protein kinase MARK2 (2.50-fold increase) involved in signal transduction, but none were associated with detoxification enzymes [44]. In CPB that had developed tolerance to a Bt biopesticide, mass spectrometry determined that antimicrobial peptides (AMP) acted as an immune response in the presence of the Bt toxins [45], rather than detoxification enzymes.

In an insecticide susceptible strain of red flour beetle (RFB) (*T. castaneum*), analyses of midgut tissues determined the greatest proteome changes associated with insecticide exposure were metabolic and antioxidant enzymes [46]. After diet exposure to diflubenzuron, 21 upregulated proteins in the midgut of RFB larvae were identified, with UDP-N-acetylglucosamine pyrophosphorylase I (a 5.3-fold increase) and glutathione synthetase (a 5.7-fold increase) showing the greatest change. The genes that were upregulated in RFB were directly associated with insecticide detoxification (P450s, GSTs, sulfotransferase, glucosyl-/glucuronosyl-transferase), ABC transporters and genes encoding cuticle structural proteins. Studies of the proteome in resistant insects have also noted changes in physiology not directly related to detoxification. A correlation between fatty acid metabolism and insecticide resistance in CPB was observed when Δ9-desaturase, which is either synthesized *de novo* or directly obtained from diet, decreased as resistance to neonicotinoid increased [13]. Transcription analysis indicated that CYP6K1 and peroxidase were also correlated with insecticide resistance as the upregulated gene was associated with fatty acid metabolism. The findings suggested that the potato beetle was either sacrificing energy storage in favour of upregulating insecticide detoxification systems or increasing fatty acid beta-oxidation for increased energy expenditure for metabolic processes.

## 5. Conclusions

In conclusion, both the multi-insecticide resistant and the tolerant CPB strains appear to have a unique but similar strategy documented in other insects to maintain higher levels of lectins that can in turn counter potential costs to immunity not observed in insecticide-susceptible CPB. This may be an adaptation to pathogens as well as toxins and demonstrates that the insect immune response shows promise as a potential target for pest management.

## Figures and Tables

**Figure 1 insects-12-00003-f001:**
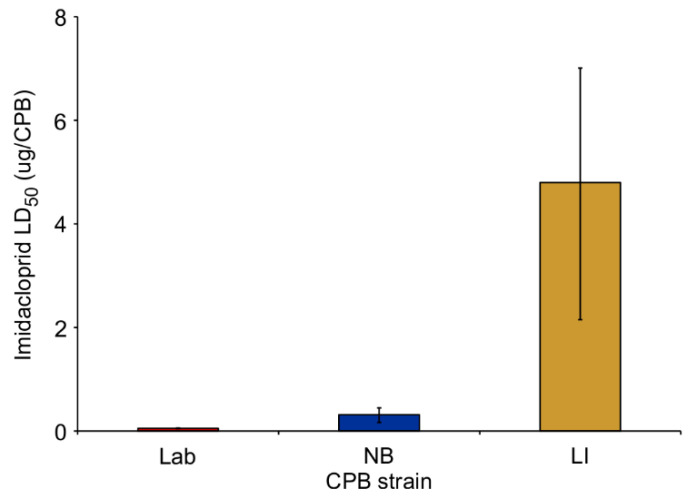
The estimated median lethal dose (LD_50_) of imidacloprid (±fiducial limits) for three Colorado potato beetle strains: laboratory insecticide-susceptible (Lab); New Brunswick field-collected strain (NB) and multi-resistant insecticide strain collected from Long Island, NY (LI) data from Scott et al. 2015 [4]).

**Figure 2 insects-12-00003-f002:**
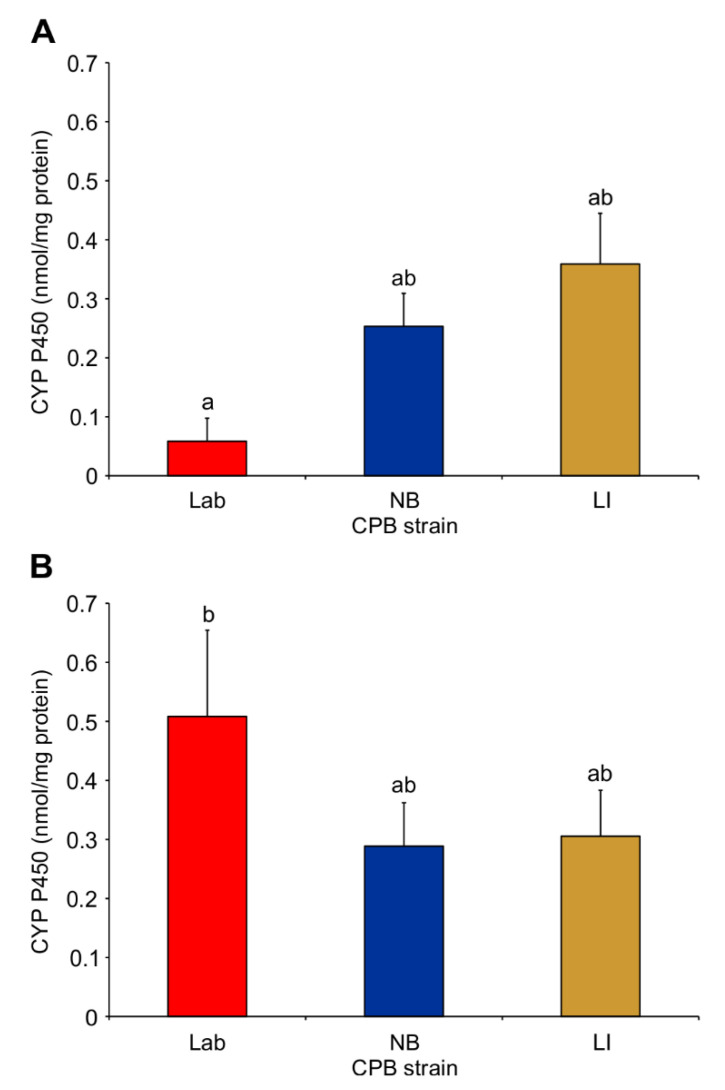
Mean cytochrome P450 activity in untreated (**A**) and imidacloprid-treated (**B**) Colorado potato beetle insecticide susceptible (Lab), tolerant (NB) and resistant (LI) strains. Bars with different lowercase letters between A and B are statistically different (Two-Way ANOVA (strain × treatment); Tukey’s pairwise comparison, *p* < 0.05).

**Figure 3 insects-12-00003-f003:**
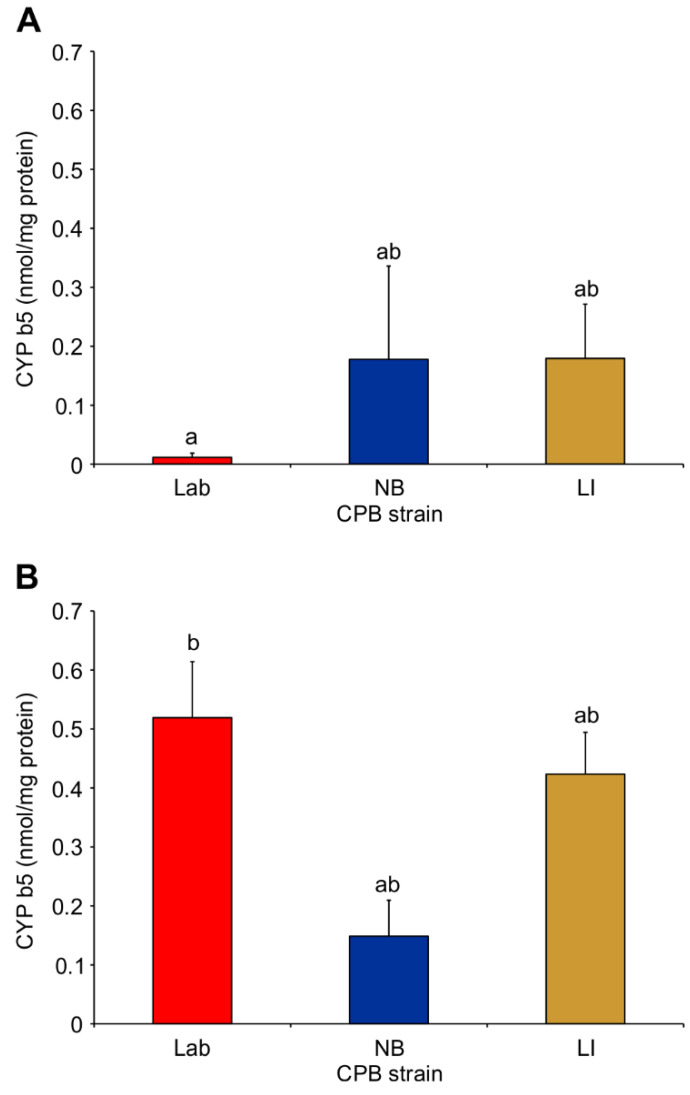
Mean cytochrome b5 activity in untreated (**A**) and imidacloprid-treated (**B**) Colorado potato beetle insecticide susceptible (Lab), tolerant (NB) and resistant (LI) strains. Bars with different lowercase letters between A and B are statistically different (Two-Way ANOVA (strain × treatment); Tukey’s pairwise comparison, *p* < 0.05).

**Figure 4 insects-12-00003-f004:**
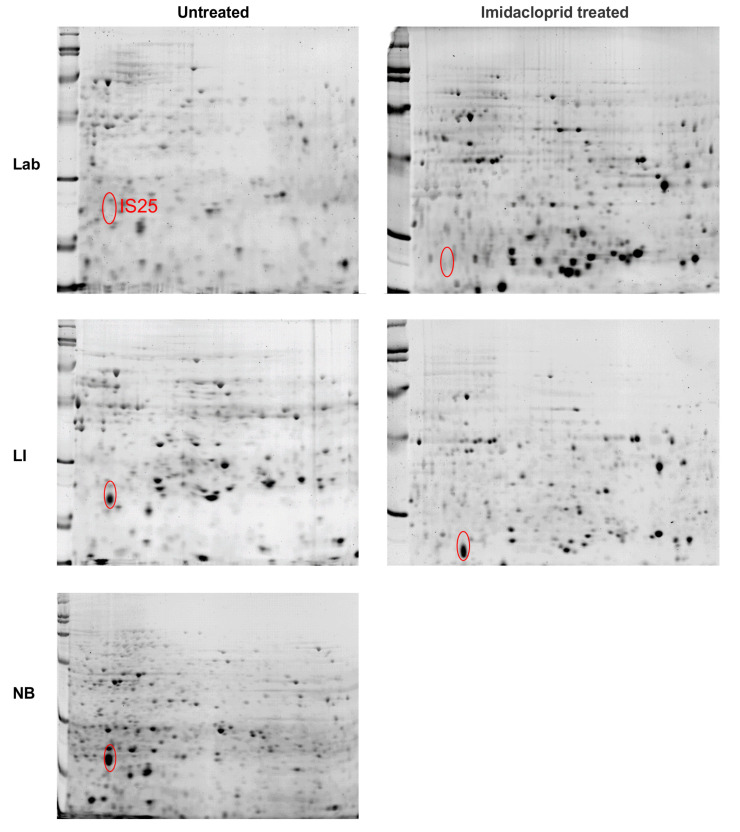
2-dimensional polyacrylamide gel electrophoresis (2D PAGE) gel separations of adult Colorado potato beetle abdominal content (minus midgut) proteins from insecticide susceptible (Lab), resistant (LI) and tolerant (NB) strains. Strips 13 cm in length were run at pH 4–7 and cut to 7cm. The location of one high concentration protein spot (IS25) in the resistant and tolerant strains (LI and NB) was marked in the gels for each strain, either pre-treated with imidacloprid or untreated.

**Figure 5 insects-12-00003-f005:**
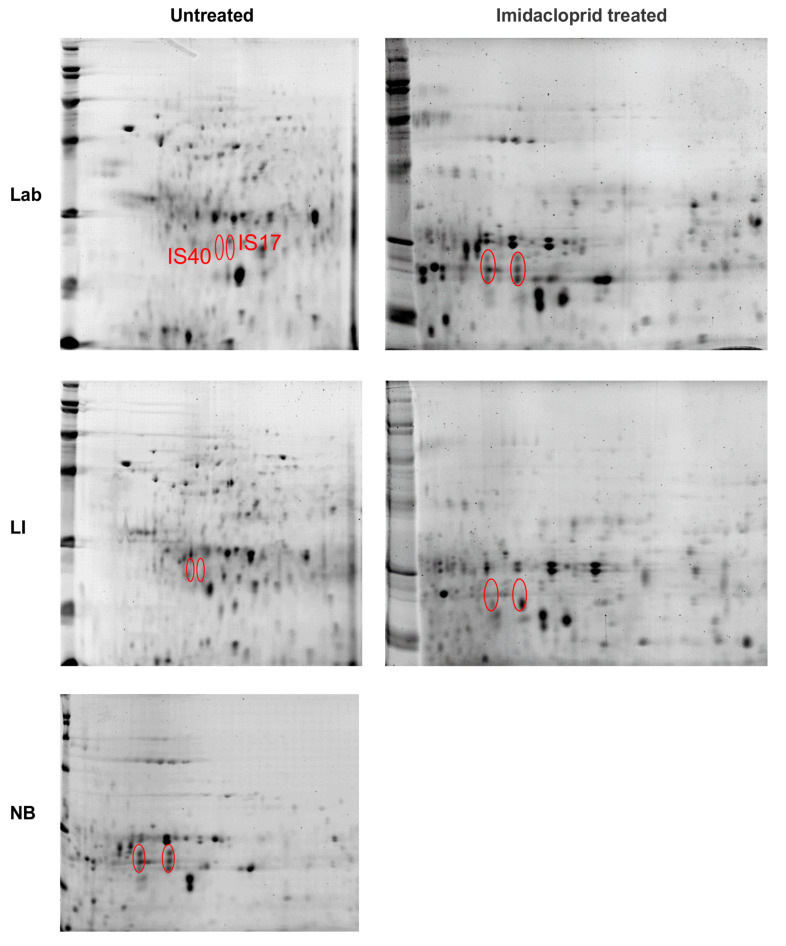
2D PAGE gel separations of adult Colorado potato beetle midgut tissue proteins from insecticide susceptible (Lab), resistant (LI) and tolerant (NB) strains. Strips 13 cm in length were run at pH 4–7 and cut to 7 cm. The location of two high concentration protein spots (IS17 and IS40) in the tolerant strain (NB) was marked in the gels for each strain, either pre-treated with imidacloprid or untreated.

**Figure 6 insects-12-00003-f006:**
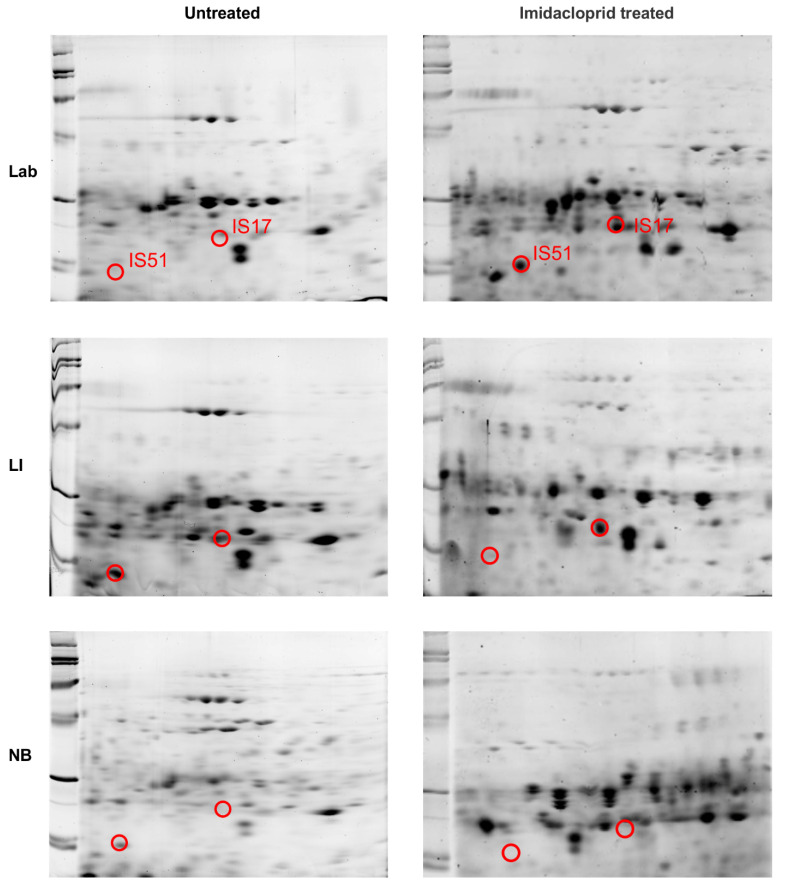
2D PAGE gel separations of adult Colorado potato beetle midgut tissue proteins from insecticide susceptible (Lab), resistant (LI) and tolerant (NB) strains. Strips 18 cm in length were run at pH 4–7 and cut to 7 cm. The location of two high concentration protein spots (IS17 and IS51) in the resistant strain (LI) were marked in the gels for each strain, either pre-treated with imidacloprid or untreated.

**Table 1 insects-12-00003-t001:** PEAKS identification searches of protein spot IS25 from Colorado potato beetle (CPB) abdominal contents, pH 4–7, 13 cm strip (cut to 7 cm).

Digestion Type	Hit #	Accession	Score −10lgP	Percent Coverage	Average Mass (Da)	^1^ Protein Description
Trypsin	1	gi|1285030099	317.44	62	19,715	ladderlectin-like
	2	gi|1285031125	209.63	31	19,830	E-selectin-like
	3	gi|1285019956	70.14	14	12,120	glutamine synthetase-like
AspN	1	gi|1285030099	300.21	56	19,715	ladderlectin-like
	2	gi|1285031125	110.03	26	19,830	E-selectin-like
	3	gi|1285029928	54.49	9	18,910	chromobox protein homolog 1-like
Chymotrypsin	1	gi|1285030099	77.46	16	19,715	ladderlectin-like

^1^ The proteins were identified in the same species, *Leptinotarsa decemlineata.*

**Table 2 insects-12-00003-t002:** PEAKS identification searches of protein spot 17 from CPB midgut, pH 4–7, 13 cm strip (cut to 7 cm).

Digestion Type	Hit #	Accession	Score −10lgP	Percent Coverage	Average Mass (Da)	Protein Description	Species
Trypsin	1	gi|1285030099	208.61	40	19,715	ladderlectin-like	*Leptinotarsa decemlineata*
	2	gi|121531620	183.54	32	29,420	digestive cysteine protease intestain, partial	*Leptinotarsa decemlineata*
	3	gi|1285031125	175.44	33	19,830	E-selectin-like	*Leptinotarsa decemlineata*
AspN	1	gi|1285030099	203.11	37	19,715	ladderlectin-like	*Leptinotarsa decemlineata*
	2	gi|121531620	102.9	9	29,420	digestive cysteine protease intestain, partial	*Leptinotarsa decemlineata*
	3	gi|1285022454	41.51	3	53,879	dihydrolipoyl dehydrogenase, mitochondrial	*Leptinotarsa decemlineata*
Chymotrypsin	1	gi|1796093563	22.06	1	62,642	dihydrolipoyl dehydrogenase	*Stenotrophomonas maltophilia*

**Table 3 insects-12-00003-t003:** PEAKS identification searches of protein spot 51 from CPB midgut, pH 4–7, 18 cm strip.

Digestion Type	Hit #	Accession	Score −10lgP	Percent Coverage	Average Mass (Da)	^1^ Protein Description
Trypsin	1	gi|121531620	86.58	8	29,420	digestive cysteine protease intestain, partial
	2	gi|815932195	71.24	4	33,107	alpha-amylase, partial
	3	gi|1285027788	50.99	2	47,891	protein disulfide-isomerase-like, partial

^1^ The proteins were identified in the same species, *Leptinotarsa decemlineata*.

## Data Availability

The data presented in this study are available on request from the corresponding author. The data are not publicly available due to privacy restrictions.

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
