# Peer review of "Proteomic Analyses Detect Higher Expression of C-Type Lectins in Imidacloprid-Resistant Colorado Potato Beetle Leptinotarsa decemlineata Say"

_insects, 2020, doi:10.3390/insects12010003_

Round 1
Reviewer 1 Report
The manuscript investigated the potential contributions/involvement of C-type lectins in strains of the Colorado potato beetle, Leptinotarsa decemlineata Say, with differential susceptibilities to insecticides, especially the neonicotinoid imidacloprid. The manuscript is technically sound and will be of great interest for many of the readers of the INSECTS journal. Below I list some minor concerns that can be better increased before the manuscript get the acceptance level.
1) Title: It is not reflecting the major findings of the manuscript and needs to be adjusted!
2) Line 20 – CPB needs to be described here as well!
3) Lines 20-22. This is pretty much speculative, and the authors jumped from one thing to another without doing any rational connection!
4) I strongly recommend the use o color to disantangle the treatment effects (i.e., CPB strains) wherever needed (especially figure 1-3). Nowadays, the vast majority of the scientific readers do their reading by using electronic devices, and considering that the jorunals do not charge colored figures for the online version, such actions facilitate the reader's comprehension and attract much more their attention!
Author Response
Reviewer 1 Comments and author responses:
- Title: It is not reflecting the major findings of the manuscript and needs to be adjusted!
The title was revised to “Proteomic analyses detect increased C-type lectins in imidacloprid-resistant Colorado potato beetle Leptinotarsa decemlineata Say”
- Line 20 – CPB needs to be described here as well!
Added the abbreviation (CPB) after Colorado potato beetle first mention on line 15
- Lines 20-22. This is pretty much speculative, and the authors jumped from one thing to another without doing any rational connection!
Edited the line to read “The 2-D gel spot identified as a CTL was greater in the insecticide resistant CPB strain, but the CTL spot size was increased by exposure to imidacloprid in the susceptible strain”.
- I strongly recommend the use o color to disantangle the treatment effects (i.e., CPB strains) wherever needed (especially figure 1-3). Nowadays, the vast majority of the scientific readers do their reading by using electronic devices, and considering that the jorunals do not charge colored figures for the online version, such actions facilitate the reader's comprehension and attract much more their attention!
Colour was added to the bars in Figures 1, 2 and 3 as requested by the reviewer.
Reviewer 2 Report
Manuscript 1040896 by Scott et al. reports results of a proteomic analyses of responses to imidacloprid exposure in resistant and susceptible strains of the Colorado potato beetle. The study was appropriately designed and executed. The manuscript is written well and is generally easy to understand. Reported findings will be interesting to a variety of readers.
I recommend that this manuscript is published by Insects. However, there are several issues that, in my opinion, need to be addressed before publication.
Lines 47-50. Need to cite a reference to back this up. I suggest Alyokhin, A., Y. H. Chen, M. Udalov, G. Benkovskaya, and L. Lindstrom. 2013. Evolutionary considerations in potato pest management. In: Giordanengo, P., C. Vincent, and A. Alyokhin [editors]. Insect Pests of Potato: Global Perspectives on Biology and Management. Academic Press, Oxford, UK. Pp. 543-571.
Line 89 implies that there were multiple field-collected populations, while line 96 states that there was only one.
Line 101. Should be “solution of technical imidacloprid in acetone.”
Line 102. What exactly were the concentrations?
Lines 102-103. How many replications?
Line 109. Abdominal what? There are many different tissues and organs in the abdomen.
Lines 162-163. What were selection criteria?
Lines 201-204 and Fig. 2. This is confusing. What were the main factors in the ANOVA? Was the interaction between them significant? Why are there letters a and ab on the graphs if all the bars are followed by the same letter a?
Lines 210-213 and Fig. 3. The same issues as in the previous comment.
Line 220 and throughout this subsection. There is no such thing as a single abdominal tissue. It is more appropriate to talk about abdominal contents minus midgut (but presumably including hindgut).
Tables 1 and 3. The last column is unnecessary. It can be replaced with a footnote that all proteins were from L. decemlineata.
Line 278. Need to elaborate on what fitness costs are and why they are important for managing insecticide resistance.
Lines 391-393. Can overproduction of lectins be a part of a general stress response?
Author Response
Reviewer 2 Comments and author responses:
- Lines 47-50. Need to cite a reference to back this up. I suggest Alyokhin, A., Y. H. Chen, M. Udalov, G. Benkovskaya, and L. Lindstrom. 2013. Evolutionary considerations in potato pest management. In: Giordanengo, P., C. Vincent, and A. Alyokhin [editors]. Insect Pests of Potato: Global Perspectives on Biology and Management. Academic Press, Oxford, UK. Pp. 543-571.
This reference was added to the manuscript at Line 51 as requested by the reviewer.
- Line 89 implies that there were multiple field-collected populations, while line 96 states that there was only one.
Line 89 was edited to read “Differential protein levels were then compared in a field-collected CPB population identified as tolerant to neonicotinoid insecticides”. Line 96 (now line 97) was not changed.
- Line 101. Should be “solution of technical imidacloprid in acetone.”
The wording was revised as suggested (now Line 102).
- Line 102. What exactly were the concentrations?
Added “(10–150 μg/mL for susceptible, 0.1-2.5 mg/mL for tolerant, 1–17.5 mg/mL for resistant)” on Line 103.
- Lines 102-103. How many replications?
Added “where a minimum of 3 replicates bioassays were performed with 10 adults per concentration” on Lines 105-106.
- Line 109. Abdominal what? There are many different tissues and organs in the abdomen.
As was suggested in comment 10 below, the description of the CPB abdominal cavity that was dissected and prepared for enzyme analysis should be referred to as “abdominal contents minus the midgut”. This was added on Line 112.
- Lines 162-163. What were selection criteria?
The selection criteria were clarified as follows: “Spots of interest were determined by eye from the overlapping gel after staining with GelCode Blue stain followed by using Progenesis SameSpots software (Nonlinear Dynamics, Newcastle upon Tyne, UK) with a one-way ANOVA (significance level of P<0.05).” This was added on Lines 166-169.
- Lines 201-204 and Fig. 2. This is confusing. What were the main factors in the ANOVA? Was the interaction between them significant? Why are there letters a and ab on the graphs if all the bars are followed by the same letter a?
The interaction in the 2-way ANOVA for the cytochrome P450 activity of the 3 strains was between the factors strain (Lab, NB and LI strains) and treatment (imidacloprid, no imidacloprid). The full statistics are reported in the text (Lines 206-209. For clarity, the caption was edited to read “Bars with different lowercase letters between A and B are statistically different (Two-Way ANOVA (strain x treatment); Tukey’s pairwise comparison, P<0.05)” – Lines 212-214.
- Lines 210-213 and Fig. 3. The same issues as in the previous comment.
The caption for Figure 3 was also edited (Lines 221-223) as above to read “Bars with different lowercase letters between A and B are statistically different (Two-Way ANOVA (strain x treatment); Tukey’s pairwise comparison, P<0.05).
- Line 220 and throughout this subsection. There is no such thing as a single abdominal tissue. It is more appropriate to talk about abdominal contents minus midgut (but presumably including hindgut).
See previous response to comment 6 above. The subsection heading (Line 225) was edited to read “Abdominal contents”. Within the section the following lines (226-229) were also revised to read “The most prominent spot of interest (IS25) found in tissues removed from the abdomen (minus the midgut) 2D PAGE strips (pH 4-7, 13 cm) was found in the NB and LI sample, but not in the Lab strain (Fig. 4). The protein of interest appears to be of the same MW and PI in tissues from both abdominal contents and the midgut”. In the following subsection (3.3.2), the heading was edited to “Midgut” from “Midgut tissue” (Line 242), and the following sentence (Lines 250-251) was revised to read “The best match using ESI LC-MS/MS peptide sequences for IS17 was also a ladderlectin-like protein from L. decemlineata (Table 2), the same as spot IS25 from the abdominal contents (Fig. 5).”
- Tables 1 and 3. The last column is unnecessary. It can be replaced with a footnote that all proteins were from L. decemlineata.
The species column was removed in Tables 1 and 3 and a footnote was added below each Table that reads “1The proteins were identified in the same species, Leptinotarsa decemlineata.”
- Line 278. Need to elaborate on what fitness costs are and why they are important for managing insecticide resistance.
We have added a further explanation of fitness costs on Lines 283-287 – “Fitness costs that have been observed in imidacloprid-resistant CPB include reductions in fecundity, such as egg hatch success [8]. Knowing that fitness costs exist in a population allows for the potential use of refuges or the promotion of rotations or other resistance management strategies to maintain the susceptibility to imidacloprid.”
- Lines 391-393. Can overproduction of lectins be a part of a general stress response?
While it is possible that the increase in CTL is part of a stress response, we do not have evidence to support this statement. Rather, we have developed our idea of it being part of a general immune response based on the findings from the references provided. Well worth further investigation though.